# Is the Increase in Record of Skin Wounds in Hospitalized Patients in Internal Medicine Units a Side Effect of the COVID-19 Pandemic?

**DOI:** 10.3390/ijerph20032228

**Published:** 2023-01-26

**Authors:** Leticia Nieto-García, Adela Carpio-Pérez, María Teresa Moreiro-Barroso, Francisco Javier Rubio-Gil, Emilia Ruiz-Antúnez, Ainhoa Nieto-García, Montserrat Alonso-Sardón

**Affiliations:** 1School of Nursing and Physiotherapy, University of Salamanca, 37007 Salamanca, Spain; 2Internal Medicine Service, Institute for Biomedical Research of Salamanca (IBSAL), University Hospital of Salamanca, 37007 Salamanca, Spain; 3Internal Medicine Service, University Hospital of Salamanca, 37007 Salamanca, Spain; 4Health Care Quality Unit, University Hospital of Salamanca, 37007 Salamanca, Spain; 5Training, Development and Innovation Area, University Hospital of Salamanca, 37007 Salamanca, Spain; 6Preventive Medicine, Epidemiology and Public Health Area, University of Salamanca, 37007 Salamanca, Spain; 7Biomedical Research Institute of Salamanca-Research Centre for Tropical Diseases at the University of Salamanca (IBSAL-CIETUS), Faculty of Pharmacy, University of Salamanca, 37007 Salamanca, Spain

**Keywords:** COVID-19, internal medicine, pressure ulcer, SARS-CoV-2, wounds

## Abstract

Wound care is an important public health challenge that is present in all areas of the healthcare system, whether in hospitals, long term care institutions or in the community. We aimed to quantify the number of skin wounds reported after and during the COVID-19 pandemic. This descriptive longitudinal retrospective study compared of wound records in patients hospitalized in the internal medicine service during the first year of the COVID-19 pandemic (from 1 March 2020, to 28 February 2021) and previous-year to the outbreak (from 1 January 2019, to 31 December 2019). A sample of 1979 episodes was collected corresponding to 932 inpatients, 434 from the pre-pandemic year and 498 from the first year of COVID-19 pandemic; 147 inpatients were diagnosed with SARS-CoV-2 infection (3.2%). The percentage of wound episodes in the first year of the COVID-19 pandemic was higher than the pre-pandemic year, 17.9% (1092/6090) versus 15% (887/5906), with a significant increase in the months with the highest incidence of COVID cases. This study shows an increase in the burden of wound care during the COVID-19 pandemic, and it could be attributable to the increase in the number of patients hospitalized for SARS-CoV-2 infection in internal medicine units.

## 1. Introduction

Literature reports that chronic wounds, especially pressure injuries (PI), represent a significant epidemiological and socio-health problem worldwide in terms of pain, disability, economic cost, loss of quality-of-life for the patient, impact on families and caregivers, and increased workload for health professionals. In addition, they are a main preventable problem in hospitalized patients associated with high morbidity and mortality [1]. Hibbs described the effects of PIs as “an epidemic under the sheets”; and she warned that 95% of them could be avoided with adequate protocol and care [2]. The National Group for the Study and Advice of Pressure Ulcers and Chronic Wounds (Spanish acronym, GNEAUPP) estimates that they occur in around 8% of hospitalized patients in Spain (up to 18% in intensive care units) and over 13% in the social and healthcare environment [3]. During decades these adverse events have remained a public health concern and a key element in healthcare quality monitoring [4]; more so when the outbreak of the novel coronavirus (2019-nCoV) began in December 2019, which could have given rise to unforeseen problems of safety and quality of health care.

The coronavirus disease (COVID-19) pandemic has severely challenged all healthcare systems around the world. Specifically, the first waves of COVID-19 may have had negative consequences for wound care due to the overburdened health care system, the urgent restructuring of health services and by limiting their services to essential activities during the lockdown, relegating wound care to the background. In addition, other factors that could influence routine care must be considered, such as the limitation of face-to-face consultations and the refusal of the patient to attend appointments for fear of exposure to COVID-19.

Although recent studies have begun to discuss the collateral damage of overburdened healthcare systems and restricted access to non-essential health services during the COVID-19 pandemic [5], few studies have focused on how COVID-19 has affected the epidemiology and the pre-pandemic wound care model [6,7,8,9,10]. A new article analyzes the influence of COVID-19 on the prevalence of PIs in the Czech Republic. The nationwide analysis shows that the prevalence of PIs was higher among patients hospitalized with the SARS-CoV-2 infection than in patients without COVID-19 [11].

In this context, we deem it appropriate to support and promote research and exchange of best practices for skin wound care and prevention of wound complications. This includes collecting information on the clinical and care burden of skin wounds on diverse healthcare systems to enable us to measure frequency and distribution as a basis to generate or formulate new and future etiological hypotheses. For this reason, we strived to conduct a single institutional study of hospitalized patients who exhibited a skin wound episode during COVID-19 crisis in internal medicine units in the Salamanca University Hospital Complex (Salamanca, Spain) to quantify skin wound care episode records and identify changes in the clinical-epidemiologic profile during the first year of the COVID-19 pandemic compared to the previous year at the beginning of the outbreak.

## 2. Materials and Methods

### 2.1. Study Design

A descriptive longitudinal retrospective study was designed to quantify and examine the records and characteristics of skin wounds (staging, location, and types of wounds) in inpatients in Internal Medicine Units during the first year of the COVID-19 pandemic (from 1 March 2020, to 28 February 2021) and the year before the outbreak (from 1 January 2019, to 31 December 2019). In Spain, the first year included three waves of the COVID-19 pandemic: 1st wave (March–June 2020); 2nd wave (July–November 2020); and 3rd wave (from December-2020 to March-2021); with a total of 3,204,531 confirmed cases and 69,609 deaths over this time period (up to 1 March 2021) [RENAVE. CNE. ISCIII: https://cnecovid.isciii.es/ (accessed on 17 January 2022)]. This study followed the Strengthening the Reporting of Observational Studies in Epidemiology (STROBE) reporting guideline.

### 2.2. Setting

This research was performed at the Salamanca University Hospital Complex (Complejo Asistencial Universitario de Salamanca “CAUSA”, in Spanish), Spain. It is a public and tertiary hospital with 903 acute beds, 110 medium-long stay beds and 45 hospital medical services. It provides healthcare cover to 331.048 inhabitants [Province of Salamanca population 1 January 2020; INE: https://www.ine.es/ (accessed on 17 January 2022)].

### 2.3. Eligibility Criteria

Inclusion criteria: all adult patients (≥18-year-old) admitted for more than 24 h to internal medicine units who presented a diagnosis of wounds in nursing records during the period 1 March 2020, to 28 February 2021, and from 1 January 2019, to 31 December 2019. The episodes include the following diagnosis: PI, vascular ulcer (arterial or venous ulcer), Moisture Associated Skin Damage (MASD), neuropathic diabetes foot or neoplastic lesions.Exclusion criteria: episodes of surgical or traumatic wounds, pediatric population and hospitalization in other non-Internal Medicine Units. Episodes recorded between 1 January and 1 March 2020 were also excluded since the reports from the National Health System were severely underreporting the number of COVID-19 cases due to the new illness and not enough adequate testing, even though there must have been SARS-CoV-2 community transmission.

### 2.4. Data Collection

Two researchers reviewed the Nursing Care Reports (NCR) records of admitted subjects who had one or more wound/s during their hospital stay at the Internal Medicine Units of the Salamanca University Hospital Complex, from 1 January 2019, to 28 February 2021. Hospital admission (HA) involves staying at a hospital for at least one night or more. The care management program GACELA-Care^®^, a software that allows the registration of nursing action in the care and monitoring of the hospitalized patient, was used for data collection from NCR. This health information system uses the “episode of care” as the unit of record, defined as “the process of care for an illness or demand made by the patient, which begins with the first contact with the health services and ends with the last contact related to the specific episode”. Data were collected by date of wound records in Gacela Care. We have included all patients admitted to the internal medicine service who presented a chronic wound record in the NCR software during the period from 1 January 2019 to 28 February 2021. Subsequently, stratified and analyzed in two groups: pre-COVID-19 episodes (from 1 January 2019, to 31 December 2019) versus first year COVID-19 episodes (from 1 March 2020, to 28 February 2021). In addition, these episodes in the first year of the COVID-19 pandemic were further subdivided and compared according to principal diagnosis: COVID-19 group versus non-COVID-19 group. Next, the Clinical Discharge Reports (CDR) corresponding to the previously selected episodes collected during the pandemic were reviewed to identify the patients with a positive diagnostic test to detect the presence of active infection by SARS-CoV-2 plus variables such as admission to Intensive Care Unit (ICU) and exitus letalis. According to Health Care authorities, a total of 4534 patients were hospitalized with a diagnosis or clinical suspicion of COVID at the University Hospital of Salamanca from 1 March 2020, to 28 February 2021 [JCYL open data portal. Epidemiological situation of the coronavirus (COVID-19) in Castilla y León:https://analisis.datosabiertos.jcyl.es/pages/coronavirus/?seccion=situacion-hospitales (accessed on 5 December 2021)].

### 2.5. Statistical Analyses

Descriptive statistics were used to describe the basic features of the data in this research: numbers (n), percentages (%), mean and standard deviation (SD). The Shapiro-Wilk test was used to verify normality. In the bivariate analysis, a Chi-square (χ^2^) test was used to test differences between two categorical variables and the measured outcome was expressed as the odds ratio (OR) together with the 95% confidence interval (CI) for OR. Continuous variables were compared with Student’s t-test or Mann-Whitney test for two groups, depending on their normal or non-normal distribution. ANOVA allowed us to examine the effects of independent (categorical) variables on a dependent (continuous) variable. To test multivariate predictive models, we applied the Logistic Regression Model (Exp(B) = OR for the predictors). A *p*-value of *p* < 0.05 was considered statistically significant. All statistical tests were performed using SPSS software (Statistical Package for the Social Sciences) version 26.0.

### 2.6. Ethics Statement

The database used during the making of this report which supports its findings can be requested from the corresponding author on a reasonable inquiry. The study was approved by the Clinical Research Ethics Committee of the University Hospital of Salamanca (Code: PI 2019 03 208). All data were kept confidential and processed anonymously in accordance with the requirements of Law 3/2018 of 5 December on the Protection of Personal Data and guarantee of digital rights.

## 3. Results

### 3.1. Skin Wound Episode Records

In the first year of the COVID-19 pandemic (from 1 March 2020, to 28 February 2021), 1092 skin wound episodes were recorded in Internal Medicine Units at the Salamanca University Hospital. In the pre-pandemic year (from 1 January 2019, to 31 December 2019), 887 skin wound episodes were recorded in the same units. Of the 1092 skin wound episodes (1st year COVID-19 pandemic), 293 episodes occurred in patients with a COVID-19 diagnosis and 799 in non-COVID-19 (see flow chart of episode records, Figure 1).

The percentage of skin wound episodes in the first year of COVID-19 pandemic was higher than pre-pandemic year, 17.9% (1092/6090) versus 15% (887/5906). During the 1st wave, the percentage of skin wound episodes was similar to pre-pandemic periods (14%), but the percentage of episodes increased significantly in the 2nd and 3rd waves (20.6% and 19.8%, respectively). Table 1 summarizes all the episode records collected and the measures of frequency calculated.

Figure 2 shows graphically this temporal evolution (months) of skin wound episodes from 1 January 2019, to 28 February 2021. The percentage of skin wound episodes in hospitalized patients with a COVID-19 diagnosis in Internal Medicine Units was 4.8% (293/6090).

Figure 3 illustrates the monthly comparison of wound episodes recorded in patients with COVID-19 and non-COVID-19 diagnoses during the first year of the pandemic. We observed a significant increase in the percentage of wound episodes in patients with a COVID-19 diagnosis in those months with the highest number of hospitalizations due to SARS-CoV-2 infection; thus, data obtained in April (7.6%) and May (10.3%) (first wave) coincided with the peak of COVID-19 hospitalizations and gradually increased again after the summer (second and third waves); while in the summer months, wound episodes occurred in patients without COVID-19 diagnosis, this coincided with the decrease in SARS-CoV-2 infection hospitalizations due to confinement.

The 1979 skin wound episodes analyzed correspond to 932 patients, 434 pre-pandemic year and 498 first year of COVID-19 pandemic; 147 inpatients were diagnosed with SARS-CoV-2 infection, representing 3.2% (147/4534) of the patients hospitalized with a diagnosis or clinical suspicion of COVID at the University Hospital of Salamanca during the first year of COVID-19 pandemic.

### 3.2. Clinical and Epidemiologic Characteristics of Skin Wounds

#### 3.2.1. 1st Year COVID-19 Pandemic Versus Pre-Pandemic

Table 2 shows main characteristics of skin wounds before and during the pandemic. No statistically significant differences were observed in the demographic profile of patients with skin wound episodes (gender, *p* = 0.807; age, *p* = 0.122) or in the type (*p* = 0.216). Significant differences were observed in the stage (*p* = 0.004) and location of the wounds (*p* = 0.006).

#### 3.2.2. COVID-19 versus Non-COVID-19

Table 3 compares the clinical and epidemiological profile of skin wound episodes in patients with COVID-19 and non-COVID-19 diagnosis. The mean age of inpatients diagnosed with COVID is significantly lower (*p* < 0.001). The probability of stage I-II skin wounds was twice as high among inpatients with a COVID-19 diagnosis [OR = 1.9; 95% CI, 1.4–2.6; *p* < 0.001]. The probability of ICU stay was six times higher among hospitalized patients with a COVID-19 diagnosis [OR = 5.7; 95% CI, 3.3–9.8; *p* < 0.001]. The case fatality rate was higher in patients with a COVID-19 diagnosis (27% versus 20.7%, *p* = 0.027).

In the second wave of the pandemic, the clinical profile of wound episodes differed due to the increased burden of patients with SARS-CoV-2 infection. Thus, the multivariable logistic regression model identified the following variables as predictors for patients with COVID-19 diagnosis: stage I-II [OR = 2.2; 95% CI, 1.2–3.9; *p* = 0.007], pressure wounds [OR = 3.0; 95% CI, 1.1–8.2; *p* = 0.028], ICU stay [OR = 8.0; 95% CI, 3.1–21.0; *p* < 0.001], and death/exitus letalis [OR = 2.5; 95% CI, 1.5–4.1; *p* < 0.001].

## 4. Discussion

The impact of the COVID-19 pandemic on wound prevention and management has been evaluated in very few studies to date [12,13,14,15,16]. A study conducted in Germany [12] shows that the COVID-19 pandemic may have had a negative consequence in terms of diagnostic work-up, hospitalization and access to a primary care physician. However, this study found no significant difference on ambulatory care or wound-specific quality of life. Another study [13] highlighted among its outcomes that 76% of health professionals consider that the pandemic affected the management of wound dressing and almost 23% of patients/caregivers believed they had a wound clinic as usual.

In Spain, the fifth prevalence study of PIs and other dependence-related skin lesions in adult patients, carried out in Spain [3], estimated a mean prevalence of 7% with higher rates in services such as palliative care, intensive care and post-surgical and reanimation units. Nevertheless, the incidence/prevalence rates are very irregular depending on the region. For example, a systematic review on the prevalence of PIs in Europe shows varying data from to 4.6% to 27.2%, with a mean of 10.8% [17]. During the COVID-19 era, the combination of very high-risk patients and the changes in the structure of routine preventive care may have led to an increase in the development of chronic wounds, mainly PIs. In addition, the high demand of intensive care in patients with severe course of COVID-19 increases the risk factors and worsens the healing of chronic wounds. A Spanish study [18] shows that the incidence of PIs in the ICUs during the first wave of the COVID-19 pandemic (March–June 2020) doubled compared to the same period in 2019, and this increase is mainly due to the presence of PIs in patients with COVID-19. However, the authors cannot be certain of the factors that influence over the appearance of PIs in patients with COVID-19. The outcome of our study establishes that wound numbers have fluctuated from a steady increase from June to August 2020, hitting the highest peak in the month of August. However, the trend in the number of total all stage wounds began to decline in September. A decrease in the records is observed during the months of March, April and May 2020, which correspond to the first great wave of COVID-19, compared to the records collected during the previous year over the same months. This may be due to the fact that during the first wave, the health care system was overloaded because of a very high rate admission of patients in short periods of time, most of which required intensive care; this led to the creation of improvised ICUs sometimes staffed by professionals not so experienced in this type of ward. These factors, along with the shortage of health professionals under quarantine compared to the needs of the system, mustered a desperate health situation that favored under-registration. Furthermore, it must be added that the need of beds for COVID-19 patients, the suspension of scheduled interventions and the restructuring of scheduled consultations during the first wave also reduced the number of admissions for other chronic pathologies, often associated with predisposing factors for the PI development. When comparing wound records in the first wave with second and third waves, an increase is observed in those months with maximum peaks of hospitalization for COVID-19.

By etiology, PIs presented the highest prevalence, followed by vascular ulcers. Regarding the PI severity, the most frequent stage of the NPUAP was stage II in both periods. The skin injuries recorded in stage I are significantly greater in number during the pandemic year. The same result is found in patients with COVID-19 when comparing staging to the non-COVID-19 group. This increase may be an indicator that the monitoring, follow-up and registration protocol was active; but the high flow of hospitalization, the isolation situation, the lack of family support and the clinical course of the disease itself magnified the risk of PI development.

According to WHO, the number of people aged 80 or over is expected to triple between 2020 and 2050 to reach 426 million [19]. Specifically, in Spain the percentage of older adults currently stands at 20% and is expected to continue increasing to 36.8% in 2050 [20]. Due to prolonged life expectancy, the high number of the elderly population and its accompanying comorbidities and disabilities, the development of chronic wounds is considered an important and challenging public health issue today and in the future [21], since it worsens the prognostic, prolongs length of hospitalization, leads to an increase in mortality and represents an increase in the cost of medical services [22,23,24]. Evidence indicates that wound care generally, and chronic wound management and treatment in particular, becomes more important as the population ages, because the prevalence of chronic wounds is highly correlated with age. As some studies indicate, the profile of the elderly patient may also be a negative factor in wound healing [25]. It is also added that patients with chronic wounds have a higher risk of suffering a serious course after SARS-CoV-2 infection due to their older age and multiple comorbidities. However, age was not identified as a statistically significant risk marker in our model. The possible cause is that this study is framed in a geographical area with a very aged population and high life expectancy. The fact that these are very high-risk patients and that it was not possible to provide the usual preventive care during the first waves of the pandemic due to healthcare overload, may have led to an increase in PIs.

As the literature has shown [26,27], health professionals have battled with different support zones than usual by increasing the use of the prone position (PP) in patient with COVID-19 with the aim of improving the prognosis in patients with respiratory distress [28]. The main affected area, consistent with previous findings [3], was sacrum and heels. Significant differences are observed between patients with COVID-19 vs non-COVID-19 when analyzing both groups. Regions especially vulnerable to the prone position such as the face and ears are recorded in patients with COVID-19 compared to non-COVID-19 group.

Finally, to meet this public health challenge, all professionals involved in the patient process should assume an active role in the performance of the skin wound prevention protocol and appropriately diagnosed and treated. In order to achieve a good quality health system, it is necessary to keep healthcare updated and training in the best practice for wound prevention and management. Skin risk assessment should be performed in both community and hospitalized setting, especially in immobilized and elderly patients.

### Limitations and Strengths

Data collection was the main methodological limitation of this research, due to: (i) it is a retrospective descriptive study that fulfills the generic purpose of collecting data to describe a phenomenon that is still poorly known, but with the limitations inherent to this type of design, it cannot explain or verify the possible underlying causes; (ii) deficient or non-completion of data on the care process in hospital records as a consequence of the impact and care burden caused by the COVID-19 pandemic in the healthcare services. In addition, the limited number of publications related to our work goals did not allow us to compare and discuss our results with previous similar studies; we can only present our observations.

However, this limitation also becomes a strength, because to our knowledge, this descriptive study provides the first analysis of the situation of wound care burden during the first three waves of COVID-19 pandemic in Spain compared to previous year. The associations established in the analysis provide the basis for future analytical research and comparisons related to the impact of COVID-19 on hospital care, as well as, improving the protocols for the prevention and enhancing registration of chronic wound in potential pandemic events.

## 5. Conclusions

COVID-19 has marked a before and after in healthcare systems around the world. This study shows an increase in the burden of wound care during the first year of COVID-19 pandemic compared to the previous one, mainly from PIs, probably due to multiple factors such as elderly patients, lack of nursing professionals in quarantine or overloaded systems. Further studies are required that explore the impact of COVID-19 pandemic on chronic wounds prevalence and the risk factors influencing these findings, more so in this society that is moving towards aging and the development of comorbidities and disabilities that lead to immobilization. In this framework, skin wound prevention is an important challenge for public health.

## Figures and Tables

**Figure 1 ijerph-20-02228-f001:**
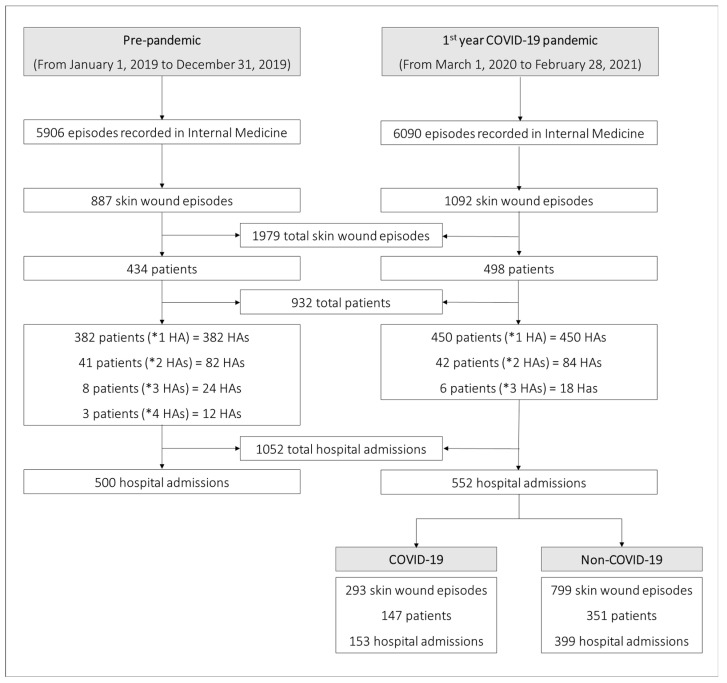
Flow chart of episode records. (* HA) refers to the number of hospital admissions.

**Figure 2 ijerph-20-02228-f002:**
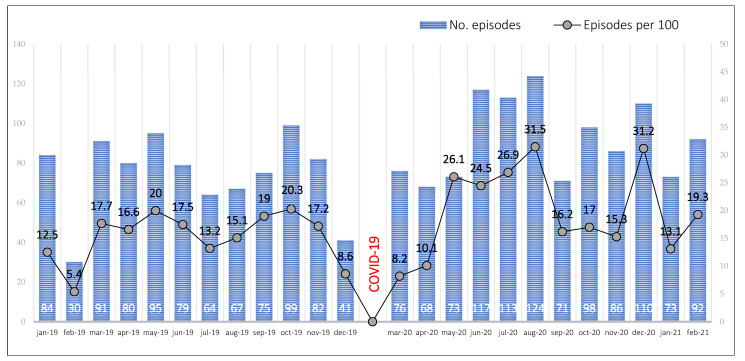
Temporal evolution of skin wound episodes before (pre-pandemic year, 2019) and during the first year of the COVID-19 pandemic in Internal Medicine Units, Salamanca (Spain).

**Figure 3 ijerph-20-02228-f003:**
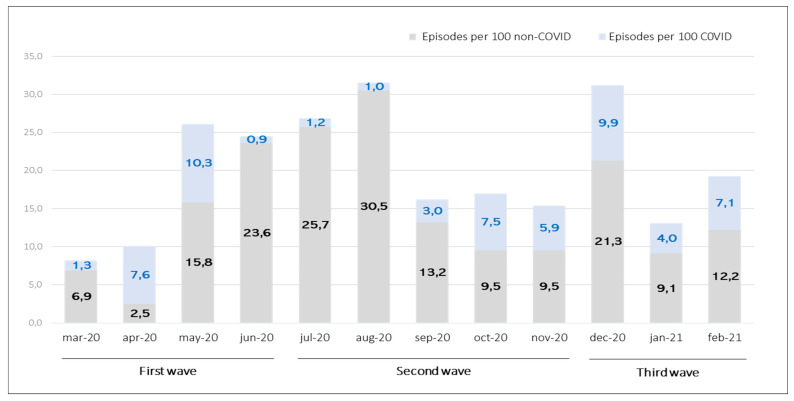
Monthly comparison of skin wound episodes (per 100) in COVID-19 versus non-COVID-19 patients during first year of the COVID-19 pandemic.

**Table 1 ijerph-20-02228-t001:** Episode registry in the C.A.U.S.A. Internal Medicine Units.

Pre-andemic Year (1 January 2019–December 2019)				First Year of the COVID-19 Pandemic (1 March 2020–28 February 2021)								COVID-19 Waves			
	TOTAL				TOTAL			No-COVID-19		COVID-19			TOTAL		
Months	Episodes	Wounds	%	Months	Episodes	Wounds	%	Wounds	%	Wounds	%	Waves	Episodes	Wounds	%
Jan-2019	672	84	12.5	Mar-2020	931	76	8.2	64	6.9	12	1.3	1st wave	2316	325	14.0
Feb-2019	556	30	5.4	Apr-2020	671	68	10.1	17	2.5	51	7.6				
Mar-2019	514	91	17.7	May-2020	367	96	26.1	58	15.8	38	10.3				
Apr-2019	480	80	16.6	Jun-2020	347	85	24.5	82	23.6	3	0.9				
May-2019	474	95	20.0	Jul-2020	420	113	26.9	108	25.7	5	1.2	2nd wave	2388	492	20.6
Jun-2019	451	79	17.5	Aug-2020	393	124	31.5	120	30.5	4	1.0				
Jul-2019	483	64	13.2	Sep-2020	439	71	16.2	58	13.2	13	3.0				
Aug-2019	443	67	15.1	Oct-2020	576	98	17.0	55	9.5	43	7.5				
Sep-2019	394	75	19.0	Nov-2020	560	86	15.4	53	9.5	33	5.9				
Oct-2019	487	99	20.3	Dec-2020	352	110	31.2	75	21.3	35	9.9	3rd wave	1386	275	19.8
Nov-2019	477	82	17.2	Jan-2021	558	73	13.1	51	9.1	22	4.0				
Dec-2019	475	41	8.6	Feb-2021	476	92	19.3	58	12.2	34	7.1				
TOTAL	5906	887	15.0	TOTAL	6090	1092	17.9	799	13.1	293	4.8	TOTAL	6090	1092	17.9

**Table 2 ijerph-20-02228-t002:** Bivariate analysis testing the main clinical and epidemiological characteristics of skin wounds during pre-pandemic year versus first year of the COVID-19 pandemic.

			Pre-Pandemic	1st Year COVID Pandemic	*p*-Value
Patients			N_1_ = 434	N_2_ = 498	*p* < 0.05
**Gender, n (%)**	Male		217 (50.0)	245 (49.2)	0.807 *
	Female		217 (50.0)	253 (50.8)	
**Age (years), mean ± SD**			84.3 ± 10.2	83.2 ± 10.6	0.122 **
**No. episodes per patient, mean ± SD**			1.7 ± 1.2	2.0 ± 1.4	0.010 **
**Episodes**			N_1_ = 887 n (%)	N_2_ = 1092 n (%)	
**Stages**	I		177 (20.0)	275 (25.3)	0.004 *
	II		403 (45.6)	478 (43.9)	
	III		206 (23.3)	198 (18.2)	
	IV		98 (11.1)	137 (12.6)	
**Types**	PI		742 (83.7)	950 (87.0)	0.216 *
	Venous ulcers		62 (7.0)	65 (6.0)	
	Arterial ulcers		13 (1.5)	16 (1.5)	
	MASD		33 (3.7)	29 (2.7)	
	Neoplastic lesions		12 (1.4)	6 (0.5)	
	Neuropathic (diabetic)		25 (2.8)	26 (2.4)	
**Anatomical positions and locations**	Supine		575 (64.8)	788 (72.2)	0.462 *
		Occiput	1 (0.1)	2 (0.2)	
		Scapulae	5 (0.6)	8 (0.7)	
		Elbows	3 (0.3)	14 (1.3)	
		Spine	16 (1.8)	28 (2.6)	
		Buttocks	42 (4.7)	49 (4.5)	
		Sacrum/coccyx	279 (31.5)	373 (34.2)	
		Heels	229 (25.8)	314 (28.8)	
	Side-lying		209 (23.6)	208 (19.0)	0.009 *
		Acromioclavicular	8 (0.9)	5 (0.5)	
		Arms/hands	3 (0.3)	16 (1.5)	
		Iliac crest	4 (0.5)	5 (0.5)	
		Trochanter	68 (7.7)	63 (5.8)	
		Leg	78 (8.8)	90 (8.2)	
		Malleolus	48 (5.4)	29 (2.7)	
	Prone		85 (9.6)	82 (7.5)	0.846 *
		Nose	4 (0.5)	2 (0.2)	
		Ears	8 (0.9)	5 (0.5)	
		Sternum	-	1 (0.1)	
		Breasts	1 (0.1)	1 (0.1)	
		Genitalia	6 (0.7)	5 (0.5)	
		Knees/tibial crest	5 (06)	7 (0.6)	
		Toes	61 (6.9)	61 (5.6)	
	Others		18 (2.0)	14 (1.3)	

* Pearson’s Chi-squared test. ** One-way ANOVA. Statistical significance (*p*-value < 0.05).

**Table 3 ijerph-20-02228-t003:** Bivariate analysis testing the main clinical and epidemiological characteristics of skin wounds during first year of the COVID-19 pandemic: COVID-19 versus non-COVID-19.

			COVID-19	Non-COVID-19	*p*-Value
Patients			N_1_ = 147	N_2_ = 351	*p* < 0.05
**Gender, n (%)**	Male		76 (51.7)	169 (48.1)	0.469 *
	Female		71 (48.3)	182 (51.9)	
**Age (years), mean ± SD**			80.5 ± 12.3	84.4 ± 9.6	<0.001 **
**No. episodes per patient, mean ± SD**			1.9 ± 1.3	2.0 ± 1.4	0.706 **
**Episodes**			N_1_ = 293 n (%)	N_2_ = 799 n (%)	
**Stages**	I		90 (30.8)	185 (23.2)	<0.001 *
	II		140 (47.9)	338 (42.5)	
	III		41 (14.0)	157 (19.7)	
	IV		21 (7.2)	116 (14.6)	
**Types**	PI		264 (90.1)	686 (85.9)	0.065
	Venous ulcers		11 (3.8)	54 (6.8)	
	Arterial ulcers		2 (0.7)	14 (1.8)	
	MASD		11 (3.8)	18 (2.3)	
	Neoplastic lesions		4 (1.4)	2 (0.3)	
	Neuropathic (diabetic)		1 (0.3)	25 (3.1)	
**Anatomical positions and locations**	Supine		234 (79.9)	554 (69.3)	0.020 *
		Occiput	2 (0.7)	-	
		Scapulae	4 (1.4)	4 (0.5)	
		Elbows	7 (2.4)	7 (0.9)	
		Spine	3 (1.0)	25 (3.1)	
		Buttocks	14 (4.8)	35 (4.4)	
		Sacrum/coccyx	117 (39.9)	256 (32.0)	
		Heels	87 (29.7)	227 (28.4)	
	Side-lying		32 (10.9)	176 (22.0)	0.012 *
		Acromioclavicular	2 (0.7)	3 (0.4)	
		Arms/hands	6 (2.0)	10 (1.3)	
		Iliac crest	-	5 (0.6)	
		Trochanter	9 (3.1)	54 (6.8)	
		Leg	15 (5.1)	75 (9.4)	
		Malleolus	-	29 (3.6)	
	Prone		19 (6.5)	63 (7.9)	<0.001 *
		Nose	2 (0.7)	-	
		Ears	5 (1.7)	-	
		Sternum	1 (0.3)	-	
		Breasts	-	1 (0.1)	
		Genitalia	-	5 (0.6)	
		Knees/tibial crest	1 (0.3)	6 (0.8)	
		Toes	10 (3.4)	51 (6.4)	
	Others		8 (2.7)	6 (0.8)	
**ICU**			41 (14.0)	22 (2.8)	<0.001 *
**Exitus letalis**			79 (27.0)	165 (20.7)	0.027 *

* Pearson’s Chi-squared test. ** One-way ANOVA. Statistical significance (*p*-value < 0.05).

## Data Availability

The data presented in this study are available on request from the corresponding author.

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
