# Peer review of "Is the Increase in Record of Skin Wounds in Hospitalized Patients in Internal Medicine Units a Side Effect of the COVID-19 Pandemic?"

_ijerph, 2023, doi:10.3390/ijerph20032228_

Round 1

Reviewer 1 Report

Dear authors,

Thank you for the opportunity to review your manuscript.

Please find comments for improvement:

abstract line 36-38: results are not clearly supporting the pandemic issue, perhaps results may be attributable to SARS-CoV-2 infection, for example, covid patients have an average of 2 wounds per patient and non-covid 2.3 wounds per patient (Figure 1), perhaps some of the non-covid patients already recovered of covid before and were hospitalized as non-covid later, yet the skin wound may be a consequence of recovered Covid, maybe that’s why there are more cases during Covid

line 97 please change to: Province of Salamanca population

line 143: Please Add Two-Way ANOVA (do you use Two-Way ANOVA? for comparisons presented in Table 2 and 3 One-Way ANOVA suffice?)

Figure 3 does not support your results, the reader may be misled that non-covid patients are much more prone to skin wounds, please think about the different presentation of results, you may also remove Figure 3 because all these results are presented in Table 1.

Figure 3 Legend: Episodes per 100 are missing

Table 3: Please change all p-values of 0.000 to <0.001

Line 217-218: Please change Exp(B) to OR

Author Response

Authors are grateful for your recommendations on a first version to improve the manuscript.

Abstract line 36-38: results are not clearly supporting the pandemic issue, perhaps results may be attributable to SARS-CoV-2 infection, for example, covid patients have an average of 2 wounds per patient and non-covid 2.3 wounds per patient (Figure 1), perhaps some of the non-covid patients already recovered of covid before and were hospitalized as non-covid later, yet the skin wound may be a consequence of recovered Covid, maybe that’s why there are more cases during Covid

RESPONSE: Although it is true that the results cannot be attributed only to the increase in patients hospitalized due to an infection, it is most certain that it can be related to this growth, because of both potential complications of comorbidities and the rise in the number of hospitalizations due to COVID.

Line 97: please change to: Province of Salamanca population.

RESPONSE: Done.

Line 143: Please Add Two-Way ANOVA (do you use Two-Way ANOVA? for comparisons presented in Table 2 and 3 One-Way ANOVA suffice?)

RESPONSE: We applied One-Way ANOVA on both tables. Our purpose was to test whether the data collected for a dependent variable were close to the common mean. This has been specified in the table text.

Figure 3 does not support your results, the reader may be misled that non-covid patients are much more prone to skin wounds, please think about the different presentation of results, you may also remove Figure 3 because all these results are presented in Table 1.

RESPONSE: Figure 3 was changed to a more readable version that does not mislead the reader. In addition, we have tried to improve figure interpretation in the text (lines 184-192).

Figure 3 Legend: Episodes per 100 are missing.

RESPONSE: Done

Table 3: Please change all p-values of 0.000 to <0.001.

RESPONSE: Done.

Line 217-218: Please change Exp(B) to OR.

RESPONSE: Done.

Reviewer 2 Report

Dear authors and editor,

The manuscript titled "Is the increase in record of skin wounds in hospitalized patients in Internal Medicine Units a side effect of the COVID-19 pandemic?" aimed to quantify the number of skin wounds reported after and during the COVID-19 pandemic.

There are many minor issues I'd like the authors resolve before consider the paper ready for being published in International Journal of Environmental Research and Public Health

Abstract

1-Adequate: The abstract gathers the relevant information of the manuscript.

2-Change the keywords. Delete the words "; Wound Care.", "Chronic wounds" and  "Skin wounds". Not found in the MeSH (Medical Subject Headings). Search for new keywords

Introduction

3-Adequate: The most important concepts of the subject to be developed are identified.

Materials and Methods  

4- Study Size: Explain how the study size was arrived at. 

Results

5-Adequate. Results are clear and concise. Relevant statistical data are collected.

Discussion

6-Attached is a biblography that may be useful.

Sallustro M, Florio A. The Impact of COVID-19 Pandemic on Vascular Leg Ulcers. Int J Low Extrem Wounds. 2022 Dec;21(4):661-666. doi: 10.1177/15347346211010958. Epub 2021 Apr 28. PMID: 33909491; PMCID: PMC9646899.

Jaly I, Iyengar K, Bahl S, Hughes T, Vaishya R. Redefining diabetic foot disease management service during COVID-19 pandemic. Diabetes Metab Syndr. 2020 Sep-Oct;14(5):833-838. doi: 10.1016/j.dsx.2020.06.023. Epub 2020 Jun 11. PMID: 32540738; PMCID: PMC7289094.

Yunir E, Tarigan TJE, Iswati E, Sarumpaet A, Christabel EV, Widiyanti D, Wisnu W, Purnamasari D, Kurniawan F, Rosana M, Anestherita F, Muradi A, Tahapary DL. Characteristics of Diabetic Foot Ulcer Patients Pre- and During COVID-19 Pandemic: Lessons Learnt From a National Referral Hospital in Indonesia. J Prim Care Community Health. 2022 Jan-Dec;13:21501319221089767. doi: 10.1177/21501319221089767. PMID: 35343835; PMCID: PMC8966061.

Conclusion

7-adequate

Reference 

8-adequate

Thank you very much for allowing me to review your manuscript.

Best regards.

Author Response

Authors are grateful for your recommendations on a first version to improve the manuscript.

Abstract

1-Adequate: The abstract gathers the relevant information of the manuscript.

2-Change the keywords. Delete the words "; Wound Care.", "Chronic wounds" and "Skin wounds". Not found in the MeSH (Medical Subject Headings). Search for new keywords.

RESPONSE: We deleted these words and add new keywords found in the MeSH.

Introduction

3-Adequate: The most important concepts of the subject to be developed are identified.

RESPONSE: Thank you for the comment.

Materials and Methods  

4- Study Size: Explain how the study size was arrived at. 

RESPONSE: We have included all patients admitted to the internal medicine service who presented a chronic wound record in the nursing care record software during the period from January 1, 2019 to February 28, 2021. A new phrase has been added in the text clarifying this aspect (Data collection section, lines 122-124).

Results

5-Adequate. Results are clear and concise. Relevant statistical data are collected.

RESPONSE: Thank you for the comment.

Discussion

6-Attached is a biblography that may be useful.

Sallustro M, Florio A. The Impact of COVID-19 Pandemic on Vascular Leg Ulcers. Int J Low Extrem Wounds. 2022 Dec;21(4):661-666. doi: 10.1177/15347346211010958. Epub 2021 Apr 28. PMID: 33909491; PMCID: PMC9646899.

Jaly I, Iyengar K, Bahl S, Hughes T, Vaishya R. Redefining diabetic foot disease management service during COVID-19 pandemic. Diabetes Metab Syndr. 2020 Sep-Oct;14(5):833-838. doi: 10.1016/j.dsx.2020.06.023. Epub 2020 Jun 11. PMID: 32540738; PMCID: PMC7289094.

Yunir E, Tarigan TJE, Iswati E, Sarumpaet A, Christabel EV, Widiyanti D, Wisnu W, Purnamasari D, Kurniawan F, Rosana M, Anestherita F, Muradi A, Tahapary DL. Characteristics of Diabetic Foot Ulcer Patients Pre- and During COVID-19 Pandemic: Lessons Learnt From a National Referral Hospital in Indonesia. J Prim Care Community Health. 2022 Jan-Dec;13:21501319221089767. doi: 10.1177/21501319221089767. PMID: 35343835; PMCID: PMC8966061.

RESPONSE: We have added in the Introduction section.

Conclusion

7-adequate

Reference 

8-adequate
